# EDM-TTS: Efficient Dual-Stage Masked Modeling for Alignment-Free Text-to-Speech Synthesis

**Nabarun Goswami**                                          *nabarungoswami@mi.t.u-tokyo.ac.jp*
*The University of Tokyo, Japan*

**Hanqin Wang**                                                   *wang@mi.t.u-tokyo.ac.jp*
*The University of Tokyo, Japan*

**Tatsuya Harada**                                             *harada@mi.t.u-tokyo.ac.jp*
*The University of Tokyo, Japan, RIKEN, Japan*

**Reviewed on OpenReview:** *https://openreview.net/forum?id=c7vkDg558Z*

## Abstract

Tokenized speech modeling has significantly advanced zero-shot text-to-speech (TTS) capabilities. The most de facto approach involves a dual-stage process: text-to-semantic (T2S) followed by semantic-to-acoustic (S2A) generation. Several auto-regressive (AR) and non-autoregressive (NAR) methods have been explored in literature for both the stages. While AR models achieve state-of-the-art performance, its token-by-token generation causes inference inefficiencies, while NAR methods while being more efficient, require explicit alignment for upsampling intermediate representations, which constrains the model's capability for more natural prosody. To overcome these issues, we propose an **E**fficient **D**ual-stage **M**asked **TTS** (EDM-TTS) model that employs an alignment-free masked generative approach for the T2S stage that overcomes the constrains of an explicit aligner, while retaining the efficiency of NAR methods. For the S2A stage, we introduce an innovative NAR approach using a novel Injection Conformer architecture, that effectively models the conditional dependence among different acoustic quantization levels, optimized by a masked language modeling objective, enabling zero-shot speech generation. Our evaluations demonstrated not only the superior inference efficiency of EDM-TTS, but also its state-of-the-art high-quality zero-shot speech quality, naturalness and speaker similarity.

## 1 Introduction

The integration of deep learning into speech synthesis has revolutionized the field, significantly advancing the quest for more natural and intelligible computer-generated speech. This evolution has progressed from basic concatenative (Hunt & Black, 1996) and statistical parametric synthesis methods (Black et al., 2007; Tokuda et al., 2000) to sophisticated approaches based on neural networks (Shen et al., 2018; Oord et al., 2016; Ren et al., 2019). In particular, transformer-based (Vaswani et al., 2017) sequence-to-sequence models (Borsos et al., 2023a; Wang et al., 2023b; Borsos et al., 2023b; Kharitonov et al., 2023) have become central to audio synthesis research, largely due to advances in audio tokenization techniques (Zeghidour et al., 2021; Défossez et al., 2022; Kumar et al., 2024; Baevski et al., 2020; Hsu et al., 2021). These innovations simplify the representation of audio signals by transforming them into discrete token sequences, making them easier for models to process and generate while also reducing storage requirements for large-scale datasets. Significant progress has also been made in zero-shot TTS, where speech can be generated in an unseen speaker's voice using just a few seconds of sample audio. Techniques for this include speaker encoders that capture speaker characteristics (Jia et al., 2018; Casanova et al., 2022; 2024; Lee et al., 2023) and in-context learning methods (Wang et al., 2023a; Borsos et al., 2023a;b).

Audio tokenization, as mentioned above, can be broadly categorized into two types: semantic and acoustic. Semantic tokens are derived from the clustering of intermediate features of self-supervised models (Baevski et al., 2020; Hsu et al., 2021) trained on large speech corpora, primarily to enhance speech recognition. These tokens aim to encapsulate the linguistic content within the speech signal. Acoustic tokens, on the other hand, are generated through neural audio codecs that use residual vector quantization (RVQ) techniques (Zeghidour et al., 2021; Défossez et al., 2022; Kumar et al., 2024). RVQ compresses audio by sequentially quantizing the residuals from previous steps, introducing a hierarchical structure among the levels of acoustic tokens. This structure supports hybrid and hierarchical synthesis methodologies, such as AudioLM's (Borsos et al., 2023a) coarse and fine modeling stages or SoundStorm's (Borsos et al., 2023b) level-wise iterative synthesis. Furthermore, quantized audio has been directly utilized to generate speech using AR language modeling approaches, such as VALLE (Wang et al., 2023a) and XTTS (Casanova et al., 2024).

The task of aligning text with semantic tokens for text-to-semantic (T2S) generation presents significant challenges, primarily due to the non-uniform and irregular nature of text data. This irregularity makes it difficult to establish a direct correspondence between textual elements and the more structured semantic tokens derived from speech. AR models (Borsos et al., 2023a; Casanova et al., 2024) generate tokens sequentially to ensure alignment, while NAR models ((Mehta et al., 2024; Ju et al., 2024; Lee et al., 2023; Kim et al., 2021) ) often depend on complex alignment mechanisms or external alignment tools(McAuliffe et al., 2017). These approaches can introduce additional layers of complexity.

Semantic and acoustic tokens, on the other hand, are generated by models with fixed downsampling rates, making them easier to align. This has led to explorations in the semantic-to-acoustic (S2A) modeling paradigm, where both AR (Casanova et al., 2024) and NAR (Borsos et al., 2023b) models have been utilized. Although AR models provide high-quality synthesis, they are hampered by slow generation and the potential for hallucinations. NAR models, in contrast, seek to efficiently bridge the semantic-acoustic gap with faster synthesis. Hybrid approaches that combine AR modeling for coarse token generation with NAR techniques for finer token levels have also been explored (Wang et al., 2023a; Borsos et al., 2023a). However, despite their promise, NAR models face ongoing challenges in balancing inference speed with stability. For instance, SoundStorm's iterative masked token generation method encounters both inference efficiencies and training stability issues (in our reproduction), underscoring the need for innovative approaches to fully exploit NAR models' benefits.

To overcome these challenges, we introduce EDM-TTS, a dual stage, masked modeling-based NAR approach for text-to-speech synthesis. In the T2S stage, we use a straightforward strategy: concatenating text with masked semantic tokens as input to a conformer encoder, trained using a masked language modeling objective. We employ confidence-based iterative sampling during inference, which achieves robust alignments without the need for additional alignment modules. For the S2A stage, we propose the Injection Conformer, a novel architecture designed to model the hierarchical dependencies inherent in acoustic tokenization. This architecture predicts coarse tokens at intermediate layers, with subsequent layers benefiting from the previously predicted levels. The Injection Conformer not only accelerates model convergence during training through teacher forcing but also streamlines the inference process by predicting all token levels (except the first) in a single network pass. The first level is handled using the same confidence-based iterative sampling as in the T2S stage, ensuring efficient and accurate synthesis.

The key contributions of our work are as follows.

- We present a dual-stage fully NAR and alignment-free tokenized text-to-speech synthesis model, EDM-TTS, capable of achieving state-of-the-art performance on zero-shot TTS.

- Our approach significantly enhances efficiency and synthesis speed, demonstrating significant speed-up over contemporary tokenized TTS models, without sacrificing quality of synthesized speech.

- Extensive experimental validation confirms that EDM-TTS markedly improves synthesized speech quality and robustness while achieving competitive zero-shot speaker transfer performance, establishing new benchmarks in the field of speech synthesis.

These achievements underscore the impact of our research in pushing the boundaries of efficiency, fidelity, and adaptability in speech synthesis technologies.

## 2   Discussion and Differentiation from Prior Works

Recent advances in non-autoregressive (NAR) text-to-speech (TTS) have led to various approaches for efficient, high-quality speech synthesis. Notable methods include SoundStorm (Borsos et al., 2023b) and VALL-E's NAR module (Wang et al., 2023a), which model dependencies between RVQ levels. In another direction, diffusion-based TTS models such as NaturalSpeech2/3 (Shen et al., 2018; Ju et al., 2024) and Voicebox (Le et al., 2023) have emerged, offering configurable inference-time latency along with state-of-the-art synthesis quality. Additionally, FastSpeech-style models (Ren et al., 2019) have demonstrated competitive performance through explicit duration prediction utilizing explicit text-to-speech alignment.

SoundStorm employs a training masking strategy where each quantizer level is selectively masked during training, and during inference, quantizer levels are predicted sequentially via iterative sampling conditioned on previously predicted coarser levels. Although this approach captures conditional dependencies, it suffers from sparse loss computation and requires large batch sizes and multiple refinement steps, which slows convergence.

VALL-E predicts the first RVQ level autoregressively, followed by a NAR module that uses quantizer ID conditioning; each quantization level is predicted based on previous levels and a learned quantizer identifier. However, since only one quantization level is trained per iteration, this method also results in slow convergence and increased training complexity.

Diffusion-based TTS models, such as NaturalSpeech2/3 and Voicebox, rely on a stochastic iterative refinement process with configurable inference runtimes—trading off denoising quality for speed. These models typically require significantly longer training durations (e.g., NaturalSpeech3 is trained for 1M steps, and Voicebox for 600K updates) and many inference iterations (with NaturalSpeech3 about 60 forward passes, NaturalSpeech2 up to 150 steps, and Voicebox with 32 steps) to fully remove noise and achieve high-quality synthesis. Moreover, NaturalSpeech2/3 and Voicebox also require external alignment mechanisms to map text to acoustic tokens, further increasing system complexity.

FastSpeech-style models benefit from their non-iterative decoding strategy, which enables fast inference. However, while their efficiency is a strength, these models typically lack in-context modeling for zero-shot speaker adaptation. Incorporating such adaptation would likely require additional variance adaptation with speaker embeddings—an extension that is not straightforward and would necessitate significant architectural modifications.

In contrast, our work introduces a novel *Injection Conformer* module within a two-stage TTS framework. In the first stage, we adopt a FAT-MLM (Zheng et al., 2021) and MaskGit (Chang et al., 2022)-inspired iterative sampling method for the text-to-semantic conversion, eliminating the need for explicit alignment. In the second stage, the Injection Conformer explicitly models the hierarchical structure of RVQ-based acoustic tokens by injecting coarser-level predictions as intermediate conditioning for finer levels. This design enables joint optimization with teacher forcing during training. Notably, after iterative sampling of the first RVQ level and a sequential fixed number of coarse level predictions, the remaining finer levels are predicted in parallel. As a result, our method achieves faster convergence and efficient inference—even a single step of the Injection Conformer yields usable results (as shown in Table 5)—in stark contrast to the many steps required by diffusion-based models.

This highlights our method's key advantages in training efficiency, inference speed, and architectural design, while also delineating differences with related works and offers a compelling alternative that balances quality and efficiency in speech synthesis.

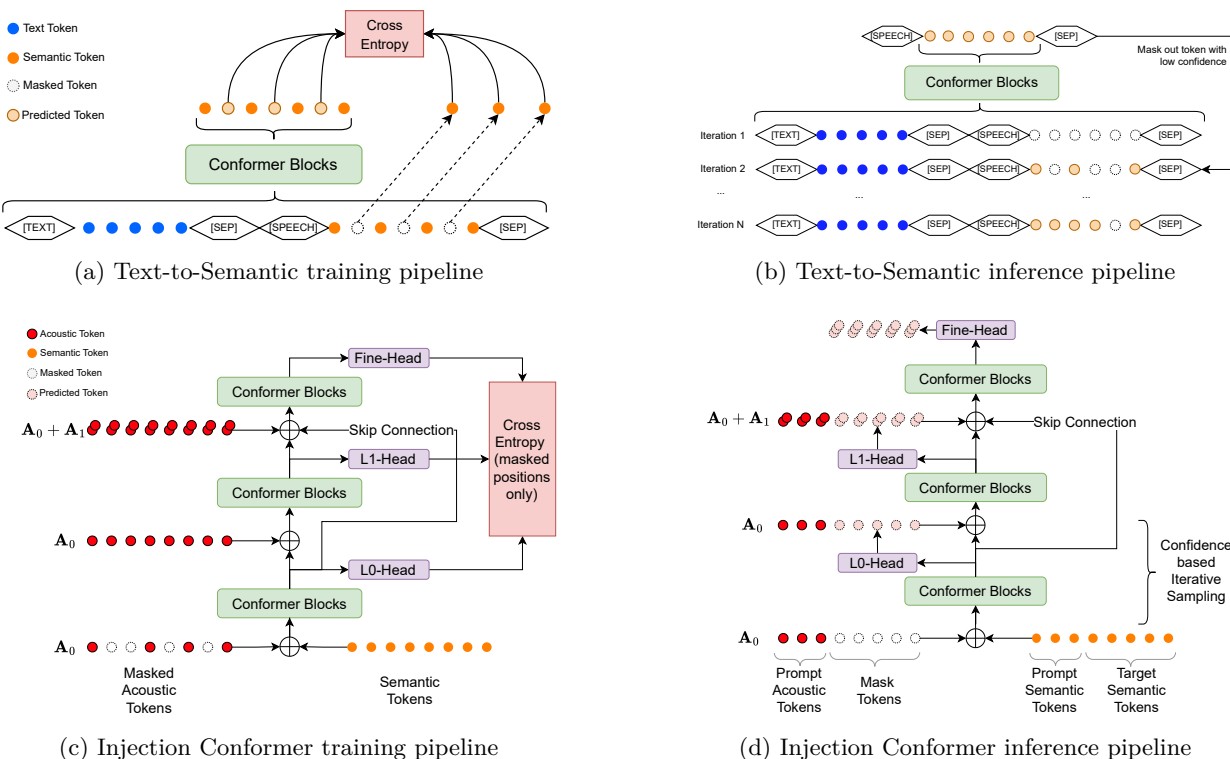

Figure 1: Model architectures and training and inference pipeline illustration. The Injection Conformer pipeline is shown with 2 injection levels for brevity.

## 3 Proposed Method

In this work, we introduce a fully non-autoregressive model that circumvents the use of complex alignment schemes, offering a more streamlined and efficient approach to text-to-semantic token generation and a new conformer-based architecture for explicitly exploiting the conditional dependence between the acoustic token levels for efficient semantic-to-acoustic token generation. The full training and inference pipelines are illustrated in Figure 1 and the following sections describe the method in details.

### 3.1 Speech and Text Tokenization

A speech signal, denoted as $x$, embodies a wealth of semantic and acoustic information. To extract semantic tokens, we employ self-supervised models that are indifferent to style, focusing instead on being closer to phonetic representations. In our work we use the HuBERT (Hsu et al., 2021) model for extracting the semantic features. These semantic tokens are identified through the assignments of clusters, utilizing a K-means model trained on the intermediate features of the model. The process can be formalized as follows:

$$\mathbf{S} = \text{KMeans}(\text{HuBERT}(x)) \tag{1}$$

where $\mathbf{S} = \{s_1, s_2, \ldots, s_m\}$ gives the semantic token sequence of the speech signal $x$. The token rate is typically much smaller than the speech sample rate.

In the domain of acoustic modeling, we leverage a neural audio codec that incorporates residual vector quantization (RVQ) to translate speech features into a succinct set of tokens. The quantization mechanism is depicted as:

$$\mathbf{F} = \text{Encoder}(x) \tag{2}$$

$$\mathbf{A}_q = \text{VQ}_q(R_q) \tag{3}$$

where $\mathbf{F}$ are the features from the encoder, $R_0 = \mathbf{F}$ and for $q > 0$, $R_q = \mathbf{F} - \sum_{i=1}^{q-1} \mathbf{A}_i$. Here, $\mathbf{A}_i$ represents the quantized tokens at the $i$-th level of quantization, and $R_q$ stands for the residual intended for the subsequent level of quantization. Through this iterative process of quantization, a hierarchical set of tokens is generated, which, when aggregated from all $Q$ quantizers, forms the acoustic token sequence $\mathbf{A}$, capturing acoustic details in an efficient manner. Given the acoustic tokens, the waveform can be recovered as illustrated below:

$$\mathbf{F}' = \sum^{Q} \mathbf{A}_q \tag{4}$$

$$\hat{x} = \text{Decoder}(\mathbf{F}') \tag{5}$$

where $\mathbf{F}'$ are the quantized features and $\hat{x}$ is the reconstructed speech signal.

For tokenizing the text inputs, we utilize a simple *utf-8* byte encoding which consists of 256 possible values for each token in the sequence. This allows us to train our model without any text pre-processing and being able to represent all characters while maintaining a small embedding layer.

### 3.2 Text-to-Semantic Modeling

Inspired by FAT-MLM (Zheng et al., 2021), we propose an alignment-free masked modeling approach where the text tokens $\mathbf{T} = \{t_1, t_2, \ldots, t_n\}$ are concatenated with the semantic tokens $\mathbf{S} = \{s_1, s_2, \ldots, s_m\}$ along the temporal dimension, $[\mathbf{T}; \mathbf{S}]$. Additionally, we insert some special marker tokens to indicate the start and end of the text and semantic token sequences as illustrated in Figure 1a. During training, we mask a proportion of the semantic tokens to obtain $\mathbf{S}_{masked}$ using a cosine schedule, inspired by SoundStorm (Borsos et al., 2023b) and MaskGIT (Chang et al., 2022), defined as

$$p(i) = \cos(\mathcal{U}(0, \frac{\pi}{2})) \tag{6}$$

where $p(i)$ is the probability of masking a semantic token at time step $i$, and $\mathcal{U}(0, \pi/2)$ is a uniform random variable between 0 and $\pi/2$. Thus, the final input to the conformer encoder is $\mathbf{X} = [\mathbf{T}; \mathbf{S}_{\text{masked}}]$.

During inference, we start with all semantic tokens masked and iteratively refine the output by sampling based on the model's confidence in its predictions. The cosine schedule is also applied during inference, modulating the number of iterations and the degree of masking. A notable benefit of our approach is the ability to control the speaking rate intrinsically. By adjusting the number of initially masked tokens, we can generate speech at varying speeds. However, this introduces the challenge of determining the optimal number of tokens to mask. To address this, we integrate a length predictor module into the model. The length predictor is a smaller conformer encoder which consumes the text tokens $\mathbf{T}$ prepended with a special learnable *length* token and applies a linear transformation to the representation corresponding to the *length* token to predict the total length $\hat{L}$ of the semantic tokens in the logarithmic domain:

$$\hat{L} = \text{Linear}(\mathbf{h}_{\text{length}}) \tag{7}$$

where $\mathbf{h}_{\text{length}}$ is the hidden state corresponding to the *length* token.

### 3.2.1 Training Objective

The primary objective of our model is to predict the masked tokens accurately. This is framed as a conditional probability problem, where we aim to model the conditional distribution $P(\mathbf{S}_{\text{masked}} \mid \mathbf{X})$. Given the model's predictions $\hat{\mathbf{S}}_{\text{masked}}$, we optimize the categorical cross-entropy loss function:

$$\mathcal{L}_{\text{t2s}} = -\sum_{i=1}^{m} \mathbf{y}_i \log \hat{\mathbf{y}}_i, \tag{8}$$

where $\mathbf{y}_i$ is the one-hot encoded ground truth semantic token for position $i$, and $\hat{\mathbf{y}}_i$ is the predicted probability distribution over the semantic token vocabulary.

For the length prediction task, the objective is to minimize the mean squared error (MSE) between the predicted length $\hat{L}$ and the logarithm of the true length $L_{\text{true}}$ of the semantic token sequence:

$$\mathcal{L}_{\text{length}} = \frac{1}{2} \left( \log L_{\text{true}} - \hat{L} \right)^2 \tag{9}$$

The total loss function for training the model is a weighted sum of the token prediction loss and the length prediction loss:

$$\mathcal{L}_{\text{total}} = \mathcal{L}_{\text{t2s}} + \lambda \mathcal{L}_{\text{length}}, \tag{10}$$

where $\lambda$ is a hyperparameter that balances the contributions of the token and length prediction tasks.

### 3.3 Semantic-to-Acoustic Modeling

#### 3.3.1 Injection Conformer Architecture

Injection Conformer comprises a sequence of conformer blocks, structured to predict acoustic tokens at coarser levels from intermediate blocks and finer level tokens from the terminal block as shown in Figure 1c. While this architecture captures the hierarchical nature of acoustic token levels, it falls short in fully leveraging the available acoustic information. To address this, we observed a key improvement area: for accurately predicting tokens at a specific quantizer level $k$, the model should have access to all pertinent information from the preceding levels. To enrich higher blocks with comprehensive acoustic information from previously predicted levels, we introduce a mechanism to inject this information directly into the subsequent blocks. This is achieved through a simple yet effective summation of the features prior to each intermediate block, formalized as follows:

$$C_{k+1} = \text{Conformer}(C_k + \text{Linear}(I_k)) \tag{11}$$

where $C_k$ represents the output from the $k^{th}$ conformer block, and $I_k$ signifies the injected information at that level. The method for calculating the injection is given by:

$$I_k = \begin{cases} \sum_0^k \mathbf{A}_q & \text{if training or prompt,} \\ \sum_0^k \hat{\mathbf{A}}_q & \text{otherwise.} \end{cases} \tag{12}$$

As an illustration, shown in Figures 1c and 1d, prediction of the L1 head is conditioned on $A_0$ only (not masked $A_1$), which is provided by the injection mechanism along with the outputs of the previous conformer block. During training, $A_0$ is derived from the ground truth tokens, while during inference it is obtained from the predictions of the L0 head.

Additionally, we also apply skip connections between the intermediate conformer blocks as shown in Figure 1c. In essence, during the training phase, the model employs ground truth acoustic tokens for computing the injections, whereas, during inference, it relies on the predicted tokens. This approach of teacher forcing not only accelerates and stabilizes training but also guides the model to focus on predicting the relevant tokens at each level by utilizing actual feature sets as a reference. This strategic injection of information ensures that each block is well-informed of the acoustic context, enhancing the model's predictive accuracy and efficiency. We set the number of coarse acoustic levels to be $K$, and the remaining $Q - K$ levels are predicted by the final conformer block. We use independent linear heads for all level outputs.

During inference phase, we prepend the prompt acoustic (first level) and semantic tokens along with fully masked acoustic tokens and target semantic tokens as input to the Injection Conformer. We then predict the first level acoustic tokens using the confidence based iterative sampling. Following this, we predict the remaining coarse levels in a single pass through the rest of the model, and using the injection mechanism as shown in Figure 1d. The last head predicts all the remaining $Q - K$ finer levels in parallel (without any explicit hierarchical modeling). For more details on the inference procedure please refer to Appendix A.2.

### 3.3.2 Training Objective

To train the Injection Conformer, we apply a Masked Language Modeling (MLM) strategy similar to the T2S stage. A portion of the first level of acoustic tokens is masked using a cosine masking schedule (Equation 6) and replaced with a learnable *mask* token. Because iterative sampling during inference is applied only for the first quantization level, higher quantization levels do not require additional masking. The input to the Injection Conformer is then constructed as the sum of the embeddings of the semantic tokens and the masked acoustic tokens from the first level:

$$\mathbf{A}_0^{\text{masked}} = \text{Mask}(\mathbf{A}_0) \tag{13}$$

$$\text{Input} = \text{Emb}(\mathbf{S}) + \text{Emb}(\mathbf{A}_0^{\text{masked}}). \tag{14}$$

The training objective is to minimize the categorical cross-entropy loss across all masked tokens for each quantization level, ensuring efficient model training and improved prediction accuracy. The loss function is given by:

$$\mathcal{L} = -\sum_{k=0}^{K} \sum_{i=1}^{N} \sum_{j=1}^{M_k} y_{i,j} \log \hat{y}_{i,j}, \tag{15}$$

where $N$ is the number of samples, $M_k$ is the number of masked tokens at level $k$, $y_{i,j}$ is the true probability distribution, and $\hat{y}_{i,j}$ is the predicted probability for token $j$ in sample $i$. This approach enables the Injection Conformer to learn effectively from the interplay of acoustic and semantic information, optimizing its ability to model complex patterns in speech data.

## 4 Experiments

In this section we first describe the training details followed by various experimental results.

### 4.1 Model Architecture and Training Details

**Training infrastructure**: We trained all models on a NVIDIA DGX-A100-80G with 8 GPUs and implement our models with the Pytorch library (Paszke et al., 2019). For efficient distributed training, we further utilized *deepspeed* (Rajbhandari et al., 2020) and performed all training with *bfloat16* (Burgess et al., 2019) percision.

**Semantic Tokenizer**: For the semantic tokenizer base model, we utilize the publicly available checkpoint [1] of the HuBERT-large model trained on the 60k hour LibriLight dataset (Kahn et al., 2020). Following the analysis of (Pasad et al., 2023), we utilize the 18th layer of the model for extracting the semantic features, which showed good overall performance in that study. We then train a 1024-cluster Kmeans tokenizer on the *train-clean-100* subset of the LibriSpeech dataset (Panayotov et al., 2015), to create the semantic tokenizer. The semantic tokenizer operates at 50hz sampling rate.

**Acoustic Tokenizer**: For the acoustic tokenizer we implement the RVQ-based codec following the architecture of (Kumar et al., 2024). We train the model with the same model architecture and optimization hyperparameters proposed by the original authors [2]. We trained the model on only speech data from the LibriLight Dataset with a batch size of 144 per GPU for 200K iterations. The acoustic tokenizer also operates at 50hz sampling rate.

**Injection Conformer**: We utilize conformer model with 16 layers with hidden size of 1024, 16 attention heads, 4096 linear size, and convolution kernel size of 5. We apply $K = 4$ injections at $[4, 7, 10, 13]$ layers. To train the model, we first pre-process the LibriLight dataset by extracting the semantic and acoustic tokens, with the above described tokenizers, for every 60 second segment. During training we randomly sample 15.36s segments, which correspond to 768 tokens at 50hz. We train with a batch size of 32 per GPU, with Adam optimizer (Kingma & Ba, 2015) with a peak learning rate of $3e^{-4}$, linearly warmed up over the first

---

[1] https://huggingface.co/facebook/hubert-large-ll60k
[2] https://github.com/descriptinc/descript-audio-codec/blob/main/conf/final/16khz.yml

4k iterations and decayed with a cosine schedule for a total of 100k iterations, which amounts to about 5 epochs over the LibriLight dataset and requiring about 12 hours. The *beta1* and *beta2* are set to $\{0.8, 0.99\}$, no weight decay, and clipped the gradient norm with a maximum of 0.5. The total number of parameters is 464M.

**Text-to-Semantic** For the text-to-semantic model, we utilize a conformer with 12 layers, hidden size of 384, 8 attention heads, 1536 linear size and convolution kernel size of 5. Additionally, we use a 4-layer conformer (other parameters are same as main model) as a length predictor, both of which are trained together. The optimizer hyperparameters are same as the Injection Conformer. To train this model, we utilize the transcripts provided by the LibriHeavy dataset (Kang et al., 2024), corresponding to 50k hours. For training efficiency we discard samples less than 1 second and larger than 25 seconds and train the model with a batch size of 32 per GPU for 300k iterations, which corresponds to about 3.5 epochs and requiring about 22 hours. For both the trainings we used bucket sampling to load the batches to reduce the amount of padding. The total number of parameters is 58M.

## 4.2 Evaluation Datasets and Metrics

### 4.2.1 Evaluation Data

To evaluate the performance of the Injection conformer on the continuation and re-synthesis tasks we utilize the 4-10s samples from the LibriTTS *test-clean* subset for a total of 1172 samples. To evaluate the out-of-domain zero-shot voice conversion performance we utilize all 18 speakers from the CMU Arctic dataset (Kominek & Black, 2004) and sample 1024 random disjoint pairs for the evaluation. We chose the CMU Arctic dataset, since none of the baselines and our method used it for training, thus enabling a fair comparison. Finally to test the zero-shot TTS performance, for the text input, we randomly sampled 40 sentences from the LibriSpeech *test-clean* subset with atleast 20 words in each sentence. While for the unseen speakers, we sampled 3-8s segments from all 20 speakers of the clean subset of the DAPS dataset (Mysore, 2014).

### 4.2.2 Automatic Metrics

We utilized several automatic metrics for the evaluation, which included UTMOS (Saeki et al., 2022) for speech quality, Speaker Encoder Cosine Similarity (SECS) using the *wavlm-base-plus-sv* model[3] for speaker similarity, Character Error Rate (CER) using the *hubert-large-ls960-ft* model[4] for percieved intelligibility and robustness, the reference-less predicted PESQ and SI-SDR metrics from the *torchaudio-squim*(Kumar et al., 2023) package for perceptual speech quality and noise levels. A point to note is that these automatic metrics are for reference and useful for quick evaluation, since it is difficult to actually judge at a fine grained level for state-of-the-art speech synthesis methods.

### 4.2.3 Subjective Evaluation

Finally we perform subjective mean opinion score (MOS) evaluation for the zero-shot TTS experiment in a comparative manner. For the subjective evaluation we use one speech sample per speaker from the zero-shot evaluation data described above. We present samples from two systems along with the text input and ask the raters to score between $\{+2, -2\}$ (order of methods is randomized), based on *naturalness, acoustic quality, prosody, and human likeness*. At least 5 independent raters rate each sample and this gives us the Comparative MOS (CMOS). Similarly we perform subjective evaluation for Speaker similarity Comparative MOS (SCMOS), the difference being for this evaluation the reference speaker prompt is also presented to the raters and asked to compare which of the two shown methods is more similar to the reference. The rating scale and number of raters per sample is same as the CMOS evaluation. For all scores, we compute the 95% confidence interval by bootstrapping. The subjective evaluation was carried out on the Amazon Mechanical Turk platform.

---

[3]https://huggingface.co/microsoft/wavlm-base-plus-sv
[4]https://huggingface.co/facebook/hubert-large-ls960-ft

### 4.3 Baselines

To compare the effectiveness of our method, we utilize several state of the art baselines across various speech generation paradigms.

- Non-iterative: HierSpeech++(Lee et al., 2023)

- Diffusion: DiffHierVC (Choi et al., 2023), StyleTTS2 (Li et al., 2024)

- Autoregressive: XTTS(Casanova et al., 2024), WhisperSpeech[5] (based on (Borsos et al., 2023a; Kharitonov et al., 2023))

- Mask-predict: SoundStorm(Borsos et al., 2023b)

We utilize publicly released checkpoints of all the above methods except SoundStorm, which we reproduce and train on the same dataset as our method and with similar parameter count. We would like to note here that training SoundStorm model is not trivial, as the training often diverges and especially at smaller batch sizes. We were only able to stably train the model to convergence with an effective batch size of 640, anything less than that did not converge in our experiments. This required multiple gradient accumulation steps to fit into the GPUs, thus increasing the the training time significantly.

WhisperSpeech also used LibriLight, with text annotations generated using an ASR model, while we used LibriHeavy annotations. StyleTTS2, XTTS, DiffHierVC, and HierSpeech++ used parts of the Libri family of datasets along with additional sources, including internal/multilingual data. While slight variations exist, we are confident that the evaluation remains fair and consistent, as we adhered to uniform experimental conditions and metrics.

Additionally, we compare with some more concurrent baselines in Appendix A.1.

### 4.4 Inference Speed Comparison

To verify the inference speed comparison, we ran three experiments. First we compare the model only runtime between SoundStorm and Injection Conformer. Both these models have the same total number of conformer blocks and similar parameter count. We measure the runtimes for voice conversion of a 60s audio with a 3s prompt, and report the average scores over 100 trials. We can see from the results in Table 1, that with the same parameter count, Injection conformer is 9x faster than SoundStorm in addition to being faster and easier to train.

Table 1: Inference (end-to-end (e2e) and model-only (mo)) runtimes for VC (60s speech) and TTS (405 characters) averaged over 100 trials on a single A100 GPU with no other running processes.

| | VC(e2e) | TTS(e2e) | VC(mo) |
|---|---|---|---|
| *non-iterative* | | | |
| HierSpeech++ | 8.56±0.15 | **0.50±0.01** | - |
| *diffusion-based* | | | |
| DiffHierVC | 8.41±0.18 | - | - |
| StyleTTS2 | - | 0.91±0.01 | - |
| *autoregressive* | | | |
| XTTS | - | 4.13±0.04 | - |
| *mask-predict* | | | |
| SoundStorm | 1.63±0.03 | - | 0.96±0.00 |
| Ours | **0.77±0.00** | 0.70±0.00 | **0.12±0.00** |

---

[5]https://github.com/collabora/WhisperSpeech

Next we performed runtime evaluations in end-to-end scenarios for both VC and zero-shot TTS. For VC we use the same setting as above, i.e. convert 60s sample with 3s prompt, while for TTS we used the *Fire and Ice* poem by Robert Frost, which consists of 405 characters and the same 3s prompt. The runtime includes prompt file reading and saving the output file to disk. Since all measurements are done on the same hardware with the same condition and averaged over 100 trials, we believe it presents a fair assessment. From the results in Table 1, we can see that the non-iterative models are the fastest, however, our method is competitive with them, while being significantly faster than diffusion and autoregressive models. A key point to note here is that HierSpeech++, while being non-iterative, for VC it needs to extract the fundamental frequencies, which causes it to significantly slow down almost to the range of diffusion based models.

## 4.5 Voice Conversion

We evaluate the performance of the Injection Conformer performance on zero-shot voice conversion. For this evaluation we utilize the CMU Arctic dataset, since none of the compared methods use it for training. We sample 1024 pairs of source and target utterances from all 18 speakers for the evaluation. The results are presented in Table 2. Injection conformer achieves the best UTMOS score while performing competitively across other metrics. Notably, when compared to SoundStorm, our method achieves better CER scores while the SECS is slightly higher for SoundStorm. Compared to diffusion based methods across both tasks, our method performs better.

Table 2: Out of distribution voice conversion on 1024 pairs from the 18 speakers of the CMU-arctic dataset

|  | UTMOS | SECS | CER | SI-SDR |
|---|---|---|---|---|
| HierSpeech++ | 4.19±0.01 | 0.81±0.01 | 2.83 | 27.58±0.17 |
| DiffHierVC | 4.03±0.02 | 0.79±0.01 | 3.81 | 26.09±0.16 |
| SoundStorm | 4.34±0.01 | **0.88±0.01** | 4.05 | 27.52±0.27 |
| Ours | **4.36±0.01** | 0.84±0.01 | **2.70** | **27.59±0.20** |

## 4.6 Zero-Shot Text to Speech Synthesis

We evaluate our method in the zero-shot TTS task against several state-of-the-art baselines as described in Section 4.3. We present the automatic metric results in Table 3. We segregate the models based on the amount of unpaired and paired training data. In terms of UTMOS and SECS, our model is at par with the current SOTA models, while achieving lowest CER among all the compared methods. We reiterate here that the automatic metrics are ok for a quick reference, however, for high quality systems, these comparisons might not reflect the actual quality.

Table 3: Zero-shot TTS performance comparison. Methods with * indicate multilingual models. UD refers to Unpaired Data while PD refers to Paired Data

|  | UD | PD | UTMOS | CER | SECS |
|---|---|---|---|---|---|
| StyleTTS2 | 94k | 245 | 4.43±0.03 | 1.59 | 0.91±0.02 |
| HierSpeech++* | 500k | 2.8k | **4.46±0.02** | 0.88 | **0.94±0.01** |
| XTTS* | - | 27k | 4.12±0.07 | 0.78 | 0.93±0.01 |
| WhisperSpeech | 60k | 60k | 3.95±0.11 | 0.66 | 0.93±0.01 |
| **Ours** | 60k | 50k | 4.43±0.02 | **0.61** | 0.91±0.01 |

Thus, to properly evaluate our method, we opt for subjective evaluations CMOS and SCMOS, as described in Section 4.2. The results of the subjective evaluation are presented in Table 4. From this evaluation we can clearly see that EDM-TTS has better speech quality and naturalness compared to SOTA methods, with all scores positive and the confidence intervals not overlapping with zero. We also performed the Wilcoxon signed-rank test (p-value) for the statistical significance of the scores. While for the SCMOS, we observed

Table 4: Comparative MOS for Speech Quality (CMOS) and Speaker Similarity (SCMOS) on a scale $\{-2, +2\}$, where positive score represents preference for the baseline methods. p-value $\leq 0.05$ indicate statistical significance.

| | CMOS (p-value) | SCMOS (p-value) |
|---|---|---|
| HierSpeech++ | -0.36 ± 0.24 (0.005) | **+0.15 ± 0.31** (0.446) |
| XTTS | -0.45 ± 0.27 (0.002) | **-0.26 ± 0.32** (0.134) |
| StyleTTS2 | -0.42 ± 0.26 (0.001) | -0.42 ± 0.29 (0.006) |
| WhisperSpeech | -0.66 ± 0.26 ($1e^{-5}$) | -0.72 ± 0.29 ($1e^{-5}$) |
| **Ours** | **0.00** | **0.00** |

that the raters marginally prefered HierSpeech++, however, the confidence interval overlaps with zero and has *p-value* $\geq 0.05$, indicating there is no significant difference between the two methods. We observed a similar trend with XTTS, albeit the raters marginally preferring our method.

## 4.7 Ablation Study

We performed an extensive ablation study to verify the effectiveness of various components of our proposed method.

### 4.7.1 Injection Conformer:

We report the results of the effects of the various architectural components in Table 5, on the reconstruction task i.e. given the first 3s as prompt and semantic tokens for the remaining, reconstruct the speech for the 4-10s samples from the LibriTTS *test-clean* subset. The top half of the table shows the results for different configurations of the model and require training under each setting, while the bottom part of the table shows the effect of number of iterations for the first level during inference. We first test the model without injections and skip connections, while still following the same multi layer outputs and objective functions. We find that even with deep supervision, the model does not perform well. Next we introduce the injections and see that as the number of injections increase the model performance also increases across all metrics. Finally arriving at our proposed configuration. All models are trained with the same hyperparameters discussed before.

Finally we evaluate the effect of increasing the number of iterations for the first level during inference. As we can see the model performs quite competitively even with a single iteration, especially in terms of CER and SECS, however, increasing the number of iterations improves the UTMOS, PESQ and SI-SDR, implying improved speech quality. We eventually arrive at 8 iterations as the gains become marginal and also gives us a good balance between fidelity and inference efficiency.

### 4.7.2 Text-to-Semantic:

The text to semantic model is quite straightforward, hence we only evaluated the various inference settings and the results are presented in Table 6. For this ablation we use the resynthesis task i.e. given first 3s as prompt and the text transcript resynthesize the speech for the 4-10s samples from the LibriTTS *test-clean* subset. First we evaluated the number of iterations during inference, we found that 16 iterations performs the best. Since our model is alignment free, it is quite robust to changes in the speech length and can seamlessly adjust the output and effectively quicken or slow down the speech rate simply by changing the desired length of the speech tokens. We see that the results of the ground truth length and more challenging settings of 0.7 and 1.3 times of the ground truth length yield similar performance across most metrics, except the CER, where the fast speech is difficult for the ASR model to transcribe, while the slow speech is comparatively easier. Finally, we applied the length predictor that was trained along with the main model and see that it achieves performance at par with using the ground truth length.

Table 5: Ablation study of different settings for the Injection Conformer (reconstruction task). For the SECS column we omit the confidence interval since it was less that the 2 decimal precision. $N$ is the number of iterations.

|  | $N$ | UTMOS | SECS | PESQ | SI-SDR | CER |
|---|---|---|---|---|---|---|
| no-inj | 8 | 3.03±0.02 | 0.90 | 2.44±0.02 | 12.10±0.2 | 1.53 |
| inj=1 | 8 | 4.10±0.02 | 0.91 | 3.36±0.02 | 20.18±0.2 | 1.27 |
| inj=2 | 8 | 4.35±0.01 | 0.91 | 3.84±0.02 | 25.26±0.2 | 1.21 |
| inj=3 | 8 | 4.40±0.01 | **0.92** | 3.92±0.01 | 26.10±0.1 | 1.20 |
| noskip | 8 | 4.42±0.01 | **0.92** | 3.98±0.01 | 27.03±0.1 | 1.18 |
| Ours | 8 | **4.43±0.01** | **0.92** | **4.03±0.01** | **27.27±0.1** | 1.15 |
|  | 4 | 4.42±0.01 | **0.92** | 4.03±0.01 | 27.17±0.1 | 1.15 |
|  | 2 | 4.39±0.01 | **0.92** | 4.00±0.01 | 26.80±0.1 | 1.15 |
|  | 1 | 4.21±0.01 | **0.92** | 3.91±0.01 | 25.26±0.2 | **1.10** |

Table 6: Ablation study of different number of iterations and length settings for the text-to-semantic (resynthesis task). For the SECS column we omit the confidence interval since it was less that the 2 decimal precision. $N$ is the number of iterations.

| $N$ | Len | UTMOS | SECS | PESQ | SI-SDR | CER |
|---|---|---|---|---|---|---|
| 16 | GT | 4.38±0.01 | 0.89 | 4.06±0.01 | 27.21±0.2 | 1.27 |
| 16 | 0.7 | 4.33±0.01 | 0.89 | 4.06±0.01 | 27.23±0.1 | 6.70 |
| 16 | 1.3 | 4.34±0.00 | 0.89 | 4.03±0.01 | 26.84±0.2 | 1.76 |
| 16 | pred | **4.40±0.01** | 0.89 | **4.07±0.01** | **27.41±0.1** | **1.20** |
| 8 | pred | 4.38±0.01 | 0.89 | 4.06±0.01 | 27.15±0.2 | 1.85 |
| 4 | pred | 4.22±0.01 | **0.90** | 4.01±0.01 | 26.64±0.2 | 7.63 |

## 5 Broader Impact and Limitations

Our proposed method achieves high-quality synthetic speech in a target speaker's voice from a short reference sample. However, as with any TTS technology, it is important to consider both the broader impact and the limitations of our approach.

On the impact side, our system may be potentially misused by malicious parties. To mitigate this risk, we have verified that our synthetic speech can be reliably detected as fake by a third-party detector[6]. Moreover, our approach does not explicitly enforce specific accents or speaking styles; rather, it adapts to the provided prompt through in-context modeling. Although this allows the model to capture speaker characteristics from a short sample, potential biases can still arise from the training data. To address this, we use the large-scale LibriLight dataset, which offers a diverse range of English speakers. While our current evaluations focus on English read-speech, future work could explore more diverse datasets to further enhance robustness across different accents, languages, and speaking styles.

Regarding computational cost and energy consumption, our method is designed to be both efficient and easy to train. By leveraging a teacher-forcing mechanism within the Injection Conformer, our model converges in only 100k iterations—substantially fewer than the over 1M iterations typically required by diffusion-based models. In addition, approaches such as SoundStorm often require larger batch sizes and longer training cycles, further increasing energy demands. Although training large-scale models inherently involves considerable computational resources, our approach offers a favorable trade-off between quality and efficiency. Future work may explore further optimizations, such as knowledge distillation or the development of lightweight model variants, to reduce these costs even more.

---

[6]https://detect.resemble.ai/

A notable limitation of our method is that the T2S model is trained with samples of up to 25 seconds in length, which may affect performance on extremely long text sequences. In practice, this limitation can be mitigated through strategies such as sentence splitting as shown in Appendix A.3.

Overall, while our work presents a significant step forward in efficient TTS synthesis, we recognize that continued research is needed to address potential biases, improve robustness across diverse linguistic conditions, and further reduce the computational footprint of state-of-the-art TTS systems.

## 6 Conclusion

In conclusion, our dual-stage fully NAR EDM-TTS model provides a solution to the challenges in both T2S and S2A modeling within speech synthesis. By utilizing a simple concatenation strategy combined with masked modeling enables us to model the T2S stage without relying on external aligners while explicitly modeling the hierarchical conditional dependencies inherent in RVQ levels, our Injection Conformer approach efficiently captures the nuanced acoustic details necessary for high-quality speech generation in the S2A stage. The architecture's ability to inject and leverage contextual acoustic information at each stage enhances both training stability and inference accuracy. This cohesive methodology not only streamlines the complex process of speech synthesis but also significantly advances the efficiency and performance of non-autoregressive models, paving the way for more natural and expressive text-to-speech systems.

**Acknowledgments**

This work was partially supported by JST Moonshot R&D Grant Number JPMJPS2011, CREST Grant Number JPMJCR2015 and Basic Research Grant (Super AI) of Institute for AI and Beyond of the University of Tokyo.

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

# A Appendix

## A.1 Concurrent Works and Comparison

As we finalized this manuscript, several recent works, such as E2-TTS (Eskimez et al., 2024), F5-TTS (Chen et al., 2024), and MaskGCT (Wang et al., 2024), have been proposed to avoid explicit alignment in non-autoregressive (NAR) TTS models. While E2-TTS and F5-TTS rely on filler tokens to match speech lengths, MaskGCT adopts a method similar to our proposed text-to-semantic stage. The emergence of these approaches highlights the growing interest and relevance of our proposed direction within the research community.

Furthermore, E2-TTS and F5-TTS utilize flow-matching-based modeling, while MaskGCT employs Sound-Storm for the semantic-to-acoustic stage. In contrast, our method introduces the Injection Conformer, an efficient semantic-to-acoustic token generation model, which distinguishes our approach from these concurrent works.

Table 7: Zero-shot TTS performance comparison with concurrent related works and closed API. Methods with * indicate multilingual models. UD refers to Unpaired Data while PD refers to Paired Data

|  | UD | PD | UTMOS | CER | SECS |
|---|---|---|---|---|---|
| Elevenlabs* | Unknown | Unknown | $4.40 \pm 0.03$ | 0.42 | $0.96 \pm 0.01$ |
| E2TTS* | - | 100k | $3.93 \pm 0.11$ | 1.08 | $0.96 \pm 0.01$ |
| F5TTS* | - | 100k | $4.15 \pm 0.09$ | 0.94 | $0.96 \pm 0.01$ |
| MaskGCT* | 100K | 100k | $3.97 \pm 0.08$ | 5.16 | $\mathbf{0.97 \pm 0.00}$ |
| **Ours** | 60k | 50k | $\mathbf{4.43 \pm 0.02}$ | **0.61** | $0.91 \pm 0.01$ |

Table 7 compares our model, EDM-TTS, with these three recent works and a popular closed API-based zero-shot TTS model, Eleven Multilingual v2 (ElevenLabs, 2025). For E2-TTS, F5-TTS, and MaskGCT, we used publicly available pre-trained checkpoints and their default settings to evaluate performance on our test set.

Notably, all three models were trained on the 100k-hour Emilia dataset, which features significantly higher speaker diversity. Consequently, it is expected for these models to achieve higher SECS scores. However, as shown in the table, our proposed EDM-TTS outperforms in terms of UTMOS (quality) and CER (character error rate), demonstrating superior audio quality and character recognizability.

Additionally, EDM-TTS achieves comparable or slightly better results than ElevenLabs Multilingual v2, despite the lack of information about the training data or model architecture for the latter. This further underscores the effectiveness and efficiency of our approach.

## A.2 Injection Conformer Inference Details

In this appendix, we provide additional details on the inference procedure of the Injection Conformer, which extends the methodology described in the main text. During inference, we first construct the input by concatenating the prompt acoustic tokens (representing the first-level acoustic tokens) with the semantic tokens, and by appending fully masked acoustic tokens corresponding to the target semantic tokens. This combined input is then fed into the Injection Conformer.

For the first quantization level, we apply a confidence-based iterative sampling procedure. A cosine-based schedule is used to gradually decrease the number of masked tokens over a fixed number of iterations. Specifically, let $T$ denote the total number of iterations and $t$ the current iteration ($t = 0, 1, \ldots, T-1$). The mask ratio $r_t$ at iteration $t$ is computed as

$$r_t = \cos\left(\frac{\pi}{2} \cdot \frac{t+1}{T}\right).$$

At each iteration, the model produces softmax probabilities $p_i$ for each token $i$. To introduce controlled randomness and enable exploration, a confidence score for each token is computed as

$$c_i = \log(p_i) + \tau \, g_i,$$

where $\tau$ is a temperature parameter (initially set to 1 and scaled by $r_t$), and $g_i$ is a noise sample drawn from a Gumbel distribution. Tokens are then sorted by their confidence scores, and a threshold $\theta$ is determined such that tokens with $c_i < \theta$ are masked in the next iteration. This iterative process, inspired by MaskGIT (Chang et al., 2022), refines the prediction of the first quantization level. This procedure is also used in the text-to-semantic inference.

For the coarse levels where the injection mechanism is applied, each level is conditioned on the predictions from the previous levels, explicitly modeling cross-level dependencies. Based on our ablation study (see Table 5), we set $K = 4$; that is, the first four levels leverage explicit cross-level conditioning through the injection mechanism. The remaining finer levels (i.e., the $Q - K$ levels) are then predicted in a single forward pass using greedy decoding, conditioned on the coarser levels. This design choice significantly improves inference speed while maintaining synthesis quality.

Furthermore, in the Injection Conformer (illustrated in Figures 1c, 1d), the L1 head is conditioned on semantic tokens and on unmasked $A_0$ provided by the injection mechanism. During training, unmasked $A_0$ is derived from the ground truth tokens, while during inference it is obtained from the predictions of the L0 head. The L1 head also benefits from features extracted by the first Conformer block, whose inputs include masked $A_0$. Because iterative sampling is applied only for the first quantization level, higher quantization levels do not require additional masking.

These inference details collectively enable the Injection Conformer to achieve efficient and accurate synthesis, as evidenced by the experimental results reported in the main text.

### A.3 Long-form Performance Evaluation

Although our model is trained with a maximum sequence length of 25 seconds, the use of rotary positional encodings enables it to generalize to slightly longer sequences without significant degradation. Nevertheless, because the length prediction is learned from the training data, the model may struggle with significantly longer inputs—particularly as text and speech tokens are concatenated along the sequence dimension.

Table 8: CER Evaluation on Long-form Speech Synthesis

| Condition | CER |
|---|---|
| 15–25 s | 1.26 |
| 25–35 s | 2.04 |
| Extreme long-form (w/o preprocessing) | 48.1 |
| Extreme long-form (w/ preprocessing) | 0.91 |

To assess long-form performance, we conducted two experiments. First, we compared performance on the LibriTTS (Zen et al., 2019) test-clean subset between samples lasting 15–25 seconds and those lasting 25–35 seconds. While the 25–35 second samples exhibited a modest increase in character error rate (CER), the overall performance remained acceptable. Second, we evaluated extreme long-form synthesis using 20 paragraphs generated by GPT-4o(OpenAI, 2024), each containing 500–1000 characters. Direct inference on these lengthy inputs resulted in a high CER; however, when a simple sentence-splitting strategy (using PySBD(Sadvilkar & Neumann, 2020) to segment the text into approximately 200-character chunks) was applied, the CER was dramatically reduced.

Table 8 summarizes the CER results under these different conditions. These findings indicate that while our model is not explicitly optimized for extremely long sequences, it remains functional for moderately longer inputs and can be effectively adapted to extreme cases through straightforward text pre-processing.

