# OpenReview forum: "EDM-TTS: Efficient Dual-Stage Masked Modeling for Alignment-Free Text-to-Speech Synthesis"
_TMLR — Accepted by TMLR_

### Review · Reviewer_5y6B · 2025-01-20

**Summary Of Contributions:**

The author(s) introduce an alignment-free, fully non-autoregressive (NAR) dual-stage text-to-speech (TTS) model.
This work targets to resolve inefficiencies of autoregressive (AR), i.e., as streaming focus or more interests.

The proposed model uses a masked generative approach inspired by FAT-MLM, eliminating the need for explicit alignment mechanisms.
Next, during a Semantic-to-Acoustic (S2A) Stage. the authors introduced Injection Conformer for multi resolution quantization.

The authors provide some evaluation on zero-shot TTS, voice conversion, and speech quality using both automatic and subjective metrics.

**Audience:**

Yes

**Broader Impact Concerns:**

- The model's performance across diverse languages, accents, and speaking styles remains unclear, which could introduce biases if not properly addressed.

- Training large-scale models, as outlined in the experimental section (DGX-A100-80G infrastructure), involves significant computational costs, raising concerns about energy consumption.

**Claims And Evidence:**

Yes

**Requested Changes:**

- Exploring long-form speech synthesis and report results would be a big plus.

- for boarder impact, provide a more detailed discussion of how prosody and expressiveness are handled in the model.

**Strengths And Weaknesses:**

### Pros

- The proposed masked modeling strategy offers a compelling alternative to traditional AR and NAR methods, avoiding complex alignment procedures while retaining high fidelity as a form of Alignment-Free TTS.

- The authors introduce Injection Conformer, which effectively models of acoustic dependencies across quantization levels and enhances training stability and inference efficiency.

### Cons

- The reliance on pre-trained models such as HuBERT and RVQ codecs introduces potential bias and dependencies that might limit the generalization of the system across diverse datasets.

- Complexity of training pipeline can be severe. Despite being NAR, the iterative confidence-based sampling and multiple quantization levels might introduce overhead in tuning and hyperparameter selection.

- The paper mentions that the model was trained with a maximum sequence length of 25 seconds. Performance degradation for longer text inputs remains unexplored.

---

> ### Author Response · Authors · 2025-02-08
> **Official Comment by Authors (1/2)**
>
> We appreciate the reviewer’s thorough feedback, recognition of our work's strengths, and constructive insights. Below, we address the main points raised.
>
> **Response to Cons**
>
> **Reliance on Pre-trained Models and Potential Bias**
> We acknowledge the reviewer’s concern regarding reliance on pre-trained models such as HuBERT and RVQ codecs. While our method builds on these self-supervised representations, this approach is widely adopted in modern TTS systems due to its ability to capture rich acoustic and phonetic features without requiring explicit supervision. Our model is trained and evaluated on LibriLight’s 60k-hour dataset, which includes a large and diverse set of English speakers, ensuring robust generalization within the English read speech domain. While we do not explicitly evaluate multiple languages or accents, our zero-shot TTS and voice conversion results demonstrate that the model generalizes effectively to unseen speakers. Additionally, the modular nature of our framework allows for easy substitution of alternative self-supervised representations, making it adaptable to different datasets if required. We will clarify these aspects in the final version to highlight the flexibility and generalization of our approach.
>
> **Training Pipeline Complexity**
> The two-stage text-to-semantic and semantic-to-acoustic paradigm is a well-established framework in modern TTS, making our overall training structure fairly standard. While tuning the number of injection layers for multi-level quantization requires some consideration, selecting the first few coarse levels as injection layers works well with minimal tuning, as demonstrated by our ablation study.
>
> Additionally, iterative sampling is only required for the first RVQ level and text-to-semantic generation, meaning that the number of quantization levels does not affect inference. During training, we employ a standard masked language modeling (MLM) approach without special conditions, ensuring stability and ease of implementation. Hyperparameter tuning (e.g., batch size, learning rate) is common across all neural models and does not introduce significant additional complexity in our case. Given these factors, we argue that our method does not impose substantially higher training complexity compared to other modern TTS frameworks.
>
> **Long-form Performance Evaluation**
> Our model is trained with a maximum sequence length of 25 seconds. However, due to the use of rotary positional encodings, it is not inherently constrained to this limit and generalizes to slightly longer sequences without significant degradation. Since length prediction is learned from training data, it may struggle with significantly longer durations, and concatenating text and speech tokens along the sequence dimension may pose challenges for extremely long inputs.
>
> To assess long-form performance, we conducted two experiments:
>
> 1. **Evaluating moderate length extensions (15-25 and 25-35 seconds)**
>    - We compared performance between 15-25s samples and 25-35s samples of the LibriTTS test clean subset. While we observed a slight degradation in CER, the performance remained reasonable and did not significantly impact usability.
>
> 2. **Evaluating extreme long-form synthesis (multi-paragraph inputs)**
>    - We tested 20 paragraphs generated by GPT-4o, each containing 500-1000 characters. Direct inference on these long inputs resulted in severe degradation, with a CER of 48.10.
>    - However, using a simple sentence-splitting strategy (PySBD) to segment paragraphs into ~200-character chunks significantly improved performance, reducing CER to 0.91.
>
> **CER Evaluation on Long-form Speech Synthesis**
>
> | CER | Condition |
> |-----------|------------|
> | 1.26 | 15-25s |
> | 2.04 | 25-35s |
>
> | CER | Condition |
> |-----------|--------------------------------|
> | 48.1 | Extreme long-form (w/o text preprocessing) |
> | 0.91 | Extreme long-form (w/ text preprocessing) |
>
> These results suggest that while our model is not explicitly optimized for long-form synthesis, it remains functional for moderately longer sequences and can be effectively adapted to extreme cases with simple text pre-processing. We will clarify these findings in the final version.

---

> > ### Author Response · Authors · 2025-02-08
> > **Official Comment by Authors (2/2)**
> >
> > **Response to Broader Impact Concerns**
> >
> > **Performance Across Diverse Languages, Accents, and Speaking Styles**
> > Our approach does not explicitly define accents or speaking styles but instead adapts to the provided prompt through in-context modeling. The model generates speech based on a short reference sample, inherently capturing speaker characteristics rather than enforcing predefined accent or style constraints.
> >
> > That said, as with any data-driven approach, potential biases may exist depending on the training data. To mitigate this, we use the large-scale LibriLight dataset, which includes a diverse range of speakers, ensuring broad coverage within the English language. While our current evaluations focus on English read-speech, future work could explore training on more diverse datasets to further enhance robustness across accents and speaking styles.
> >
> > **Computational Cost and Energy Consumption**
> > We acknowledge the reviewer’s concern regarding the computational cost of training large-scale models. However, our method is designed to be efficient and easy to train, achieving high-quality results with significantly fewer iterations compared to other state-of-the-art approaches.
> >
> > - Due to the teacher-forcing mechanism in the Injection Conformer, our model converges within 100k iterations, whereas diffusion-based models require over 1M iterations to reach comparable quality.
> > - Additionally, SoundStorm requires more than twice the training time, as it struggles to converge with smaller batch sizes, leading to longer training cycles.
> >
> > While energy consumption is a valid concern, we emphasize that our model's efficiency mitigates excessive computational overhead. By requiring fewer training iterations and leveraging a structured approach to quantization modeling, our method offers a better trade-off between quality and efficiency compared to many existing high-fidelity TTS models. Future work could explore further optimizations, such as knowledge distillation or lightweight model variants, to reduce computational costs even further.
> >
> > **Conclusion**
> > We appreciate the reviewer’s valuable feedback and constructive suggestions. Our responses clarify key aspects of our approach, including generalization, training complexity, long-form performance, and computational efficiency. We have demonstrated that our method remains competitive while being easier to train and more efficient than comparable high-quality TTS models. Additionally, our results indicate that with simple preprocessing, our model can effectively handle longer sequences. We will incorporate these clarifications in the final version and explore further improvements in future work. Thank you for your time and consideration.

---

### Review · Reviewer_G1Tv · 2025-02-02

**Summary Of Contributions:**

This paper proposes a 2-stage approach for TTS, composed of a text-to-semantic token model (T2S) and a semantic-to-acoustic token model (S2A), where the acoustic token is hierarchical, obtained from a residual vector quantization (RVQ) module.

The key novelty lies in the design of the S2A module, which the authors referred to as the Injection Conformer. This Conformer model predicts coarser tokens at earlier layers, and the prediction is fed as additional input to later layers to capture the correlation between coarse tokens and fine tokens (see Fig 1c, 1d). Both T2S and S2A are trained with a masked language modeling objective, which closely follows SoundStorm and MaskGiT. During inference, T2S and the first-level RVQ tokens are decoded with confidence-based iterative sampling while the rest are done in one pass to improve efficiency.

**Audience:**

Yes

**Claims And Evidence:**

Yes

**Requested Changes:**

1. In Sec 2.3.1, the authors put “Following this, we predict the remaining levels in a single pass through the rest of the model” Does this mean the remaining Q-K levels of acoustic tokens are predicted in parallel, indicating that correlation between these layers are not modelede?
2. In Figure 1(c), illustration of the S2A model, the L1 head is conditioned on masked A0, semantic tokens, and ground truth A0 as input. Does the model not take masked A1 as input for the second set of Conformer blocks?
3. Please discuss the difference between VALL-E NAR module and SoundStorm in detail, in terms of what correlation (inter quantization layer, inter frame within a level) are modeled and what the relationships are.

**Strengths And Weaknesses:**

Strengths
1. This paper proposed a fully NAR token-based TTS model based on a two-stage MLM design. NAR enables parallel decoding which improves the efficiency compared to AR based approaches like VALL-E or hybrid approaches like SoundStorm.
2. Specifically the S2A module can predict RVQ tokens of all level with a single model in one forward pass, in contrast to the NAR decoder of VALL-E and SoundStorm which requires at least Q (number of quantization levels) passes to the model, because those models are trained to model $P(A_{q+1} | S, A_{1..q}) \forall q$, where $S$ denotes semantic tokens and $A_q$ denotes $q$-th level acoustic token
3. Experimental results show better performance than baseline methods, especially SoundStorm which is the most related


Weaknesses
1. The paper is missing a crucial related work section to provide a complete discussion of how the proposed method differs from prior work. In particular, the design differences compared to the VALL-E NAR module, SoundStorm S2A module, FastSpeech2 style NAR models should be thoroughly discussed. The authors are also missing discussion with diffusion style TTS models such as NaturalSpeech2/3, Voicebox, which is another type of prominent NAR models that enjoys configurable inference-time latency
2. The proposed S2A is highly related to VALL-E and SoundStorm NAR modules. Both prior work also model inter-layer correlation, since $A_{q+1}$ are conditioned on both S and $A_{1…q}$, and “injection” is also done in these work since there inputs are $\sum_{i=1}^q E_i(A_i)$ where E is an embedder. The S2A component can be interpreted having separate but jointly optimized models for $P(A_{q+1} | S, A_{1..q})$ while SoundStorm and VALL-E have shared $P(A_{q+1} | S, A_{1..q})$ for all $q$.

---

> ### Author Response · Authors · 2025-02-09
> **Official Comment by Authors (1/2)**
>
> We sincerely appreciate the reviewer’s thoughtful feedback and recognition of our work’s contributions. The detailed insights and constructive suggestions have helped us clarify key aspects of our approach. Below, we address each concern and provide additional explanations where necessary.
>
> **Response to Weaknesses**
>
> 1. **Missing Related Work Section**
>
> We acknowledge the reviewer’s concern regarding the lack of a dedicated related work section. In the revised manuscript, we will include a more detailed discussion comparing our method to prior works, including VALL-E NAR, SoundStorm, FastSpeech2-style NAR models, and diffusion-based TTS models such as NaturalSpeech2/3 and Voicebox. We will highlight key differences in training methodology, inference efficiency, and modeling of inter-quantization dependencies, particularly emphasizing the advantages of the Injection Conformer and parallel prediction strategy.
>
> 2. **Comparison of Our S2A Component to SoundStorm and VALL-E NAR**
>
> While SoundStorm and VALL-E NAR also model inter-layer correlations, our approach differs in how these dependencies are captured, leading to faster training convergence and more efficient inference.
>
> - **SoundStorm** models inter-quantization level dependencies through a training masking strategy, selectively masking one quantizer level while keeping coarser levels unmasked and masking all finer levels. At inference, quantizer levels are predicted sequentially via iterative sampling. This approach leads to inefficient training due to sparse loss computation and requires large batch sizes to stabilize. Our reproduction of SoundStorm showed that it fails to converge with smaller batch sizes, requiring a batch size of 640 with gradient accumulation, significantly increasing training time.
>
> - **VALL-E NAR** conditions each quantization level on the previous levels using a quantizer ID embedding. However, only one quantizer level is trained per iteration, leading to slower convergence. The VALL-E paper reports a training duration of 800k iterations, significantly higher than our method.
>
> - **Our Injection Conformer** explicitly models inter-quantization dependencies at the architectural level. This allows for:
>   - Jointly optimizing all quantizer levels per iteration, computing loss for all masked positions simultaneously.
>   - Capturing hierarchical dependencies through injected coarse-level predictions.
>   - Enabling teacher forcing during training, leading to faster convergence and more stable optimization.
>
> Regarding the reviewer’s interpretation that our method has separate but jointly optimized models for $ P(A_{q+1} | S, A_1..q) $ while SoundStorm and VALL-E use a shared $P(A_{q+1} | S, A_1..q) $, this is partially correct. Our approach can be viewed as an unrolled version of their shared model formulation, but with smaller blocks per level and more direct intermediate conditioning via injection. This enables efficient training and inference while maintaining hierarchical dependencies.
>
> For the final finer quantization levels, we predict them in parallel instead of injecting coarser predictions further. Our ablation study (Table 5) shows that injecting beyond levels 3-4 does not significantly improve performance, so further injections would add complexity without meaningful gains. Thus, beyond this threshold, all finer levels are predicted in parallel, balancing efficiency and modeling power.

---

> > ### Author Response · Authors · 2025-02-09
> > **Official Comment by Authors (2/2)**
> >
> > **Response to Requested Changes**
> >
> > 1. **Clarification on Prediction of Remaining $Q-K $ Levels (Sec 2.3.1)**
> >
> > For the coarse levels where injection is applied, each level is conditioned on the previously predicted levels following the injection-based prediction mechanism, explicitly modeling dependencies.
> >
> > For the remaining finer levels ($Q-K $), they are predicted in parallel. However, their correlation is not completely unmodeled. As shown in our ablation study (Table 5), adding more injection levels beyond 3-4 results in diminishing performance gains. Based on this, we set $ K=4 $ in our proposed method, meaning the first four levels leverage explicit cross-level correlation via injection, while the remaining levels are predicted in parallel, conditioned on these coarser levels. This balances efficiency and performance, as additional injections do not significantly improve synthesis quality but increase computational overhead.
> >
> > Additionally, iterative sampling is applied only for the first quantization level to ensure a refined and confident prediction at this stage. All remaining levels, including injected coarse levels and the final parallel-predicted finer levels, are decoded greedily, significantly improving inference speed without sacrificing quality.
> >
> > We will clarify this in the final version to ensure an accurate understanding of our modeling approach.
> >
> > 2. **Clarification on Figure 1(c) and Conditioning of L1 Head**
> >
> > The reviewer is correct that the L1 head is conditioned on semantic tokens and $ A_0 $, but it does not use masked $ A_1 $ directly. Instead, the injection mechanism provides unmasked $ A_0 $ as input:
> > - **During training**, this comes from the ground truth tokens.
> > - **During inference**, this is based on predictions from the L0 head.
> >
> > Additionally, the L1 head uses features from the first Conformer block, whose inputs include masked $ A_0 $. Since iterative sampling is applied only for the first quantization level ($ A_0 $), higher levels do not require masked versions. The L1 head predicts based on semantic tokens, injected $ A_0 $, and processed features from the first Conformer block, ensuring hierarchical information flow without redundant masking at higher quantization levels.
> >
> > We will clarify this in the final version to ensure the figure and description accurately reflect this conditioning mechanism.
> >
> > 3. **Discussion on Differences Between VALL-E NAR and SoundStorm**
> >
> > We appreciate the reviewer’s request for further discussion on how VALL-E NAR and SoundStorm model inter-quantization layer and intra-frame correlations. We have already addressed this in detail in our response to Weakness 2. We will further refine these explanations and add them in the revised manuscript ensure clarity.
> >
> > **Conclusion**
> >
> > We sincerely appreciate the reviewer’s insightful feedback and constructive suggestions. We have addressed concerns regarding model design, inference strategy, and comparisons to prior works and will incorporate the requested related work discussion in the revised manuscript. Additionally, we have clarified our modeling choices, particularly regarding inter-level dependencies, parallel prediction, and training efficiency. These refinements will strengthen the clarity and impact of our work. Thank you for your time and consideration.

---

### Review · Reviewer_QAq3 · 2025-02-08

**Summary Of Contributions:**

The paper under review presents a Text-to-Speech (TTS) model that claims to achieve efficient NAR speech synthesis and voice converison. While the authors demonstrate some promising experimental results, the overall contribution of the paper is limited, and several critical aspects are lacking, which significantly impacts the paper's scientific rigor and reproducibility.

**Audience:**

Yes

**Claims And Evidence:**

No

**Requested Changes:**

Enhance Novelty: The authors should clearly articulate the unique contributions of their work and, if possible, introduce more innovative elements to distinguish their model from existing approaches.

Expand Comparative Analysis: The paper should include comparisons with a broader range of recent TTS models to better demonstrate the proposed model's performance and relevance.

Provide Reproducibility Resources: It is imperative that the authors release the code, model, and audio samples. This will not only enhance the paper's credibility but also allow the community to build upon the work.

Elaborate on Key Techniques: The authors should provide a detailed explanation of the confidence-based iterative sampling method used in the Injection Conformer module. This should include its implementation, rationale, and its impact on the model's performance. Such details are crucial for readers to fully understand and evaluate the proposed approach.

**Strengths And Weaknesses:**

Strengths:

Efficient Parallel Synthesis: The paper highlights the model's ability to synthesize speech non-autoregressively, which is a notable feature in modern TTS systems. The reported efficiency gains, if validated, could be of interest to the community.

Experimental Results: The authors provide experimental results that show competitive performance in terms of synthesis speed and speech quality. These results suggest that the model is functional and achieves reasonable performance on the chosen benchmarks.

Aligner-Free TTS: The proposed model adopts an aligner-free approach, which aligns with recent trends in the TTS community. Recent works such as SeedTTS, F5-TTS, and DiTTo-TTS have also explored aligner-free architectures, demonstrating their potential to simplify the TTS pipeline and improve efficiency. The authors' exploration of this direction is commendable and contributes to the ongoing evolution of TTS systems. This approach is not only relevant but also meaningful for the broader TTS community, as it reflects a shift toward more streamlined and scalable architectures.

Weaknesses:

Lack of Novelty: The core components and methodologies employed in the proposed model appear to be largely derived from prior work.  The paper does not introduce significant innovations or novel techniques, which limits its contribution to the field. The authors should more clearly delineate their specific contributions and differentiate their work from existing approaches.

Insufficient Comparative Analysis: While the paper presents experimental results, it fails to compare the proposed model against many state-of-the-art TTS systems. This omission makes it difficult to assess the true advancement or competitiveness of the proposed approach. A more comprehensive comparison with recent models (such as SPEAR-TTS, Voicebox, DITTO-TTS, F5-TTS) is necessary to contextualize the results.

Lack of Reproducibility: One of the most significant shortcomings of this paper is the absence of essential resources for reproducibility. The authors do not provide access to the model's code, trained weights, or synthesized audio samples. In the field of TTS, providing audio samples is considered a fundamental requirement, as it allows the community to evaluate the perceptual quality of the generated speech. Without these resources, the paper's claims cannot be independently verified, which undermines its scientific value.

Insufficient Detail on Key Techniques: The authors mention the use of confidence-based iterative sampling in the context of the Injection Conformer module, which appears to be a critical component of the model. However, they provide very little detail about how this technique is implemented or its impact on the generated results. Given the potential importance of this method for both the performance and understanding of the model, a more thorough explanation is necessary. This omission leaves readers with an incomplete understanding of the proposed approach and its contributions.

---

> ### Author Response · Authors · 2025-02-11
> **Official Comment by Authors (1/2)**
>
> We sincerely appreciate the reviewer’s thoughtful feedback and constructive suggestions. In response, we have carefully addressed each of your concerns, providing detailed clarifications on our contributions, comparative analysis, reproducibility resources, and key technical techniques.
>
> **Response to Weaknesses**
> 1. **Lack of Novelty**
>
> We acknowledge that a two-stage text-to-semantic and semantic-to-acoustic token-based TTS paradigm has been explored in prior work. However, our approach introduces several key innovations. First, in the text-to-semantic stage we adopt a MaskGit-style iterative modeling technique that removes the need for explicit alignment between text and semantic tokens—a departure from many previous NAR models. (It is worth noting that concurrent works such as E2-TTS, F5-TTS, and Mask-GCT have also explored similar directions.) Second, in the semantic-to-acoustic stage we propose a novel model, the Injection Conformer that explicitly captures the hierarchical structure of RVQ-based acoustic tokens through our innovative injection mechanism. This mechanism enables the use of teacher forcing during training, leading to both faster convergence and stable training, as well as efficient inference and state-of-the-art performance.
>
> 2. **Insufficient Comparative Analysis**
>
> We compare our method against several state-of-the-art TTS and VC approaches. While SPEAR-TTS and Voicebox are not open source and thus difficult to evaluate directly, we compare with Whisper-Speech—an open-source variant built on the SPEAR-TTS concept, which is cited in Sec 3.3. Additionally, Ditto-TTS and F5-TTS, we believe are concurrent works; in Appendix A1, we include comparisons with three concurrent models (E2-TTS, F5-TTS, and Mask-GCT) as well as a widely used closed-source API (ElevenLabs TTS). Our results show that our method achieves better performance in terms of speech quality (UTMOS) and CER, though it is slightly lower in SECS. This discrepancy is expected since the concurrent models were trained on the 100k-hour Emilia dataset, which features significantly higher speaker diversity, likely contributing to higher SECS scores. We believe these comparisons adequately contextualize our results and demonstrate the competitiveness of our approach.
>
>
> 3. **Lack of Reproducibility**
>
> We appreciate the reviewer’s focus on reproducibility. However, we would like to respectfully note that synthesized audio samples were provided in the supplementary material (please refer to the zip file and open the index.html for easy access). Moreover, while the current submission does not include our code and trained weights, we are fully committed to reproducibility. In this rebuttal, we provide an anonymized code link, and the final version of the manuscript will include the complete GitHub repository. We hope that these measures will address the concern and allow for independent verification of our results.
>
> [Anonymized code link](https://zenodo.org/records/14838315?preview=1&token=eyJhbGciOiJIUzUxMiJ9.eyJpZCI6IjAyMTgyMGIyLWI1MjgtNDlmNi1hZDhlLTJkNGQ4YWNjNjYwNiIsImRhdGEiOnt9LCJyYW5kb20iOiJlNTBiNThkNDNhOGNjZWM2NWRlZDJhODkzYTI3ZTMwOSJ9.Oa84LuZNg5qqkGLPcjiDf13tQuQyP28K6zYF_Q9uxSSxnSp0aZOXxxjZ7CG3L6ABx5lIVV8-AeAXYbD2pyN0Sw)

---

> > ### Author Response · Authors · 2025-02-11
> > **Official Comment by Authors (2/2)**
> >
> > 4. **Insufficient Detail on Key Techniques**
> >
> > We appreciate the reviewer’s request for additional details on our confidence-based iterative sampling method, which is critical to the performance of our Injection Conformer module. We now provide a more detailed explanation, including its mathematical formulation.
> >
> > During inference, we adopt a cosine-based schedule to determine the mask ratio at each iteration. Let $T$ be the total number of iterations and $t$ denote the current iteration ($t = 0, 1, \ldots, T-1$). The mask ratio $r_t$ is computed as:
> > $$
> > r_t = \cos\left(\frac{\pi}{2} \cdot \frac{t+1}{T}\right).
> > $$
> > This schedule gradually decreases the number of masked tokens as inference proceeds.
> >
> > For each token prediction, the model outputs softmax probabilities $p_i$ for each token $i$. To incorporate randomness and enable exploration, we compute a confidence score for each token:
> > $$
> > c_i = \log(p_i) + \tau \, g_i,
> > $$
> > where $\tau$ is a temperature parameter, and $g_i$ is a noise sample drawn from a Gumbel distribution. The temperature starts at 1 and is gradually decreased by scaling with $r_t$. We then sort the tokens by their confidence scores and determine a threshold corresponding to the top-$k$ tokens, where $k$ is determined by the desired number of tokens to mask for that particular iteration. Specifically, if $\theta$ is the $k$-th smallest confidence value, token $i$ is masked for the next iteration if:
> > $$
> > c_i < \theta.
> > $$
> >
> > This iterative process, inspired by MaskGIT [1], refines the predictions by gradually unmasking tokens. We have shared an anonymized code link above with full implementation details.
> >
> > We believe this additional explanation clarifies our confidence-based iterative sampling method. The impact of the number of iterations is shown in the ablation studies in Tables 5 and 6, while for more rigorous analysis of the masking schedules we refer the readers to MaskGIT [1] paper.
> >
> > For more clarity, we will include this explanation in the updated manuscript.
> >
> > [1] Chang, H., Zhang, H., Jiang, L., Liu, C., & Freeman, W. T. (2022). Maskgit: Masked generative image transformer. In Proceedings of the IEEE/CVF Conference on Computer Vision and Pattern Recognition (pp. 11315-11325).
> >
> >
> > **Response to Requested Changes**
> >
> > We appreciate the reviewer’s suggestions to enhance novelty, expand the comparative analysis, provide reproducibility resources, and elaborate on key techniques. We would like to note that our responses to the weaknesses above already address these points in detail. In summary:
> >
> > - For enhancing novelty, we have clearly delineated our unique contributions—specifically, the MaskGit-inspired iterative modeling in the text-to-semantic stage and the novel Injection Conformer in the semantic-to-acoustic stage that leverages teacher forcing for efficient training and inference.
> > - For comparative analysis, our manuscript includes comparisons with several state-of-the-art methods (including concurrent works).
> > - Regarding reproducibility, we have included an anonymized code link in this rebuttal and will provide the complete GitHub repository in the final version. Synthesized audio samples are already available in the supplementary material.
> > - For key techniques, we have now provided an in-depth explanation and mathematical formulation of our confidence-based iterative sampling method.
> >
> > These details will be incorporated into the updated manuscript to ensure a comprehensive presentation of our contributions.
> >
> > **Conclusion**
> >
> > In summary, we believe these clarifications and updates address the reviewer’s concerns and strengthen our manuscript. We are committed to incorporating all these details into the final version, and we thank you for your time and consideration.

---

### Author Response · Authors · 2025-02-11
**Summary of Changes**

We would like to thank the reviewers for taking the time and effort to review our paper and provide invaluable feedback which has helped us improve the manuscript.

We have updated the manuscript with the following changes as requested by the reviewers:

- Added **Section 2: Discussion and Differentiation from Prior Works**, requested by reviewers **QAq3** and **G1Tv**.
- Added **Appendix A.2 Injection Conformer Inference Details**, requested by reviewers **QAq3** and **G1Tv**.
- Added **Appendix A.3 Long-form Performance Evaluation**, requested by reviewer **5y6B**.
- Updated **Section 5: Broader Impact and Limitations** to include a more in-depth discussion on handling specific accents, speaking styles and computational cost and energy consumption for training large scale models. Requested by reviewer **5y6B**.

All changes in the paper are marked in blue text for easy visibility.

---

### Decision · Action_Editor_4AyA · 2025-03-17

**Recommendation:** Accept with minor revision

**Comment:**

- Featured Certification: N/A
The initial submission lacked clarity, strong comparisons, and reproducibility. While the authors provided reasonable rebuttals addressing most concerns, the reviewers still left concerns on the significance.
Despite the lack of the novelty, the reviewers concur "leaning toward accept" by its merits.

- Reproducibility: No.
Initially, the code and trained weights were missing and the reviewers left concerns, but they were later provided and promised in the rebuttal.

- Survey certification: N/A

**Audience:**

The paper contributes to the speech synthesis community, focusing on aligner-free, non-autoregressive architectures.

**Claims And Evidence:**

Key claims

- The paper introduces an efficient, fully non-autoregressive TTS model. In particular, Injection Conformer for semantic-to-acoustic transformation effectively models hierarchical dependencies across quantization levels.
- Despite the hierarchical dependencies, the model achieves efficient parallel synthesis, reducing inference time compared to AR-based and hybrid TTS approaches.

The reviewers were concerned about the initial submission due to the lack of comparison and validation to support the claims.
The reproducibility was another concern.
The authors later addressed these by comparing and contrasting it with the existing models during the revision.
Also, by providing the code and weight, the reproducibility was also resolved during the revision.
The reviewers turned to "leaning toward accept" with the majority of their concerns resolved during the discussion period.

The commitment with further refinements and explicit integration of rebuttal clarifications into the final version is required.

**Unclearly reflected points requests by G1Tv**
- Clarification on Prediction of Remaining
 Levels (Sec 2.3.1)
- Clarification on Figure 1(c) and Conditioning of L1 Head

The authors promised to clarify these in the final version. This should be reflected in the revision.

---

> ### Author Response · Authors · 2025-04-06
> **Author Response to Minor Revisions (Camera-Ready)**
>
> Thank you for your efforts in overseeing the review process of our paper.
>
> Regarding the minor revisions requested, we had previously included the relevant clarifications in Appendix A2. For the camera-ready version, we have now integrated both of the requested clarifications directly into the main text, specifically in Section 3.3.1 (which was Section 2.3.1 in the original submission—the section number has changed due to the addition of a new section earlier in the paper).
>
> Additionally, we have included a link to the GitHub repository containing the code, to further support reproducibility.
>
> We hope these updates satisfactorily address the remaining concerns. We would like to sincerely thank the Action Editor and reviewers for their valuable feedback, which has greatly improved the quality of our manuscript.